# *Ampelomyces* strains isolated from diverse powdery mildew hosts in Japan: Their phylogeny and mycoparasitic activity, including timing and quantifying mycoparasitism of *Pseudoidium neolycopersici* on tomato

Márk Z. Németh[1◐], Yuusaku Mizuno[2◐], Hiroki Kobayashi[2], Diána Seress[1], Naruki Shishido[2], Yutaka Kimura[2], Susumu Takamatsu[3], Tomoko Suzuki[4], Yoshihiro Takikawa[5], Koji Kakutani[6], Yoshinori Matsuda[2], Levente Kiss[1,7]*, Teruo Nonomura[2,8]*

1 Centre for Agricultural Research, Plant Protection Institute, Eötvös Loránd Research Network, Budapest, Hungary, 2 Laboratory of Phytoprotection, Science and Technology, Faculty of Agriculture, Kindai University, Nara, Japan, 3 Faculty of Bioresources, Mie University, Mie, Japan, 4 Department of Chemical Biological Sciences, Faculty of Science, Japan Women's University, Tokyo, Japan, 5 Plant Center, Institute of Advanced Technology, Kindai University, Wakayama, Japan, 6 Pharmaceutical Research and Technology Institute, Kindai University, Osaka, Japan, 7 Centre for Crop Health, University of Southern Queensland, Toowoomba, Australia, 8 Agricultural Technology and Innovation Research Institute, Kindai University, Nara, Japan

◐ These authors contributed equally to this work.
* nonomura@nara.kindai.ac.jp (TN); Levente.Kiss@usq.edu.au (LK)

## Abstract

A total of 26 *Ampelomyces* strains were isolated from mycelia of six different powdery mildew species that naturally infected their host plants in Japan. These were characterized based on morphological characteristics and sequences of ribosomal DNA internal transcribed spacer (rDNA-ITS) regions and actin gene (*ACT*) fragments. Collected strains represented six different genotypes and were accommodated in three different clades of the genus *Ampelomyces*. Morphology of the strains agreed with that of other *Ampelomyces* strains, but none of the examined characters were associated with any groups identified in the genetic analysis. Five powdery mildew species were inoculated with eight selected *Ampelomyces* strains to study their mycoparasitic activity. In the inoculation experiments, all *Ampelomyces* strains successfully infected all tested powdery mildew species, and showed no significant differences in their mycoparasitic activity as determined by the number of *Ampelomyces* pycnidia developed in powdery mildew colonies. The mycoparasitic interaction between the eight selected *Ampelomyces* strains and the tomato powdery mildew fungus (*Pseudoidium neolycopersici* strain KTP-03) was studied experimentally in the laboratory using digital microscopic technologies. It was documented that the spores of the mycoparasites germinated on tomato leaves and their hyphae penetrated the hyphae of *Ps. neolycopersici*. *Ampelomyces* hyphae continued their growth internally, which initiated the

**Data Availability Statement:** All nucleotide sequences are available from the DNA Data Bank of Japan nucleotide sequence database (https://www.ddbj.nig.ac.jp/index-e.html) (accession numbers MT645844 to MT645849, MT644626 to MT644651 and MT656304 to MT656329). The other relevant data are within the paper and its Supporting Information files.

**Funding:** This work was partly supported by Grants for Scientific Research from Faculty of Agriculture, Kindai University, and Agricultural Technology and Innovation Research Institute, Kindai University. MZ Németh's stay in Japan was supported by the Institute of Fermentation, Osaka. The funders had no role in study design, data collection and analysis, decision to publish, or preparation of the manuscript.

**Competing interests:** The authors have declared that no competing interests exist.

atrophy of the powdery mildew conidiophores 5 days post inoculation (dpi); caused atrophy 6 dpi; and complete collapse of the parasitized conidiophores 7 dpi. *Ampelomyces* strains produced new intracellular pycnidia in *Ps. neolycopersici* conidiophores *ca*. 8–10 dpi, when *Ps. neolycopersici* hyphae were successfully destroyed by the mycoparasitic strain. Mature pycnidia released spores *ca*. 10–14 dpi, which became the sources of subsequent infections of the intact powdery mildew hyphae. Mature pycnidia contained each *ca*. 200 to 1,500 spores depending on the mycohost species and *Ampelomyces* strain. This is the first detailed analysis of *Ampelomyces* strains isolated in Japan, and the first timing and quantification of mycoparasitism of *Ps. neolycopersici* on tomato by phylogenetically diverse *Ampelomyces* strains using digital microscopic technologies. The developed model system is useful for future biocontrol and ecological studies on *Ampelomyces* mycoparasites.

## Introduction

Powdery mildew fungi (*Erysiphaceae*), obligate biotrophic pathogens of over 10,000 host plant species, including important crops, can cause serious losses in agriculture, horticulture and forestry [1–4]. To control powdery mildew, fungicides are regularly applied. Frequent and inadequate use of fungicides may lead to the emergence of fungicide resistance in some powdery mildew fungi [5–7]. Fungicide use may also have negative side effects on plant physiology [8] and biodiversity [9]. To mitigate these problems, new, alternative control strategies have been tested to control powdery mildews. Physical and biological methods have been proposed against powdery mildews to complement or substitute chemical control [10]. For example, mycoparasites have been successfully used against powdery mildews as biocontrol agents (BCAs). Such BCAs include *Aphanocladium album* [11], *Pseudozyma flocculosa*, *Moesziomyces rugulosus* [12, 13], *Gjaerumia minor* [14, 15], *Lecanicillium lecanii* [13] and *Ampelomyces quisqualis* [16, 17]. *Ampelomyces* spp. are slow-growing pycnidial fungi, which are widely distributed and well known mycoparasites or hyperparasites of powdery mildew fungi on many cultivated and wild plants [18–20]. Several studies have shown that *Ampelomyces* strains belong to genetically distinct lineages based on sequences of the nuclear ribosomal DNA internal transcribed spacer (rDNA-ITS) regions and actin gene (*ACT*) fragments [21–23]. Although these linages are thought to represent distinct species of the genus *Ampelomyces* [24–26] and differences between colony morphology of strains belonging to distinct clades were also mentioned [22], a taxonomic revision of the genus has not been performed yet [27].

Cross-inoculation experiments, including *in vitro* studies [25, 28–30] and field experiments [27, 31] involving different *Ampelomyces* strains and several powdery mildew species have consequently shown that these mycoparasitic strains do not show strict host specificity, i.e. they are capable of infecting many host species irrespective of the original host. This lack of specificity [10] allows BCAs composed of a single *Ampelomyces* strain to be applied against a wide range of powdery mildew species. Indeed, several studies showed the biocontrol potential of *Ampelomyces* spp. against powdery mildews on various crops, such as *Erysiphe trifoliorum* on red clover [32], *Podosphaera leucotricha* on apple [33, 34], *Podosphaera xanthii* on cucumber [22, 28, 34–40] and melon [41, 42], *Erysiphe necator* on grapevine [18, 22, 43, 44], *Blumeria graminis* on barley [45, 46] and wheat [46], and several other species, as well [22, 28, 32, 34, 40, 47–50]. Some *Ampelomyces* strains have been developed as commercial anti-powdery mildew biofungicide products such as AQ10® (Ecogen Incorporated, USA), Q-fect® (Green Biotech, Korea) and Powderycare® (AgriLife, India) [20, 26, 51]. Efficacy of biocontrol, however, varied significantly between

the experiments with *Ampelomyces*, some reporting satisfactory results [52, 53], others suboptimal control [35, 54]. These contradictions might result from experimental differences, such as humidity [20] but might also reflect differences in the mycoparasitic activities of *Ampelomyces* strains belonging to distinct lineages [20, 30], and/or presence of physiological differences between genetically similar or uniform strains of *Ampelomyces* [22, 27, 30].

For the effective utilization of *Ampelomyces* strains for biocontrol purposes, detailed knowledge is needed not only on the host range, but also on other aspects of their biology. Improved visualization was considered as an ideal approach for such studies [55]. In a previous study aimed at a better visualization of the interaction between the plant and the tomato powdery mildew fungus, *Pseudoidium neolycopersici*, we examined leaf type I trichomes of common tomato (cv. Moneymaker). These trichomes are 1.5–2.5 mm in length on a multicellular base with a small glandular tip, and abundant and longer than other types of trichomes [56]. Powdery mildew development was followed on the trichomes using a high-fidelity digital microscope [57]. Thus, from the methodological point of view, trichomes were excellent sites for analyzing the interactions between plant cells and pathogens under natural environmental conditions. Based on our previous experience [57], we assumed that it is also possible to visualize the tritrophic interactions among mycoparasites, mycohosts and plant cells in more detail under the same conditions mentioned above.

Our main aims in the present study were to 1) reveal morphological and physiological characteristics and phylogenetic placement of *Ampelomyces* strains originating from Japan; 2) test and quantify the rate of mycoparasitism in different mycohost species; and 3) observe the infection process and pycnidiogenesis of *Ampelomyces* strains in the tomato powdery mildew fungus in detail. To our knowledge, this is the first study to provide detailed analysis of Japanese *Ampelomyces* strains in terms of their phylogeny, morphology, physiology and rate of mycoparasitism. Also, this is the first study that consecutively monitored the interaction between selected *Ampelomyces* strains and powdery mildews in detail, using digital microscopic techniques, and provided quantitative data to characterize this specific type of mycoparasitic activity.

## Materials and methods

### Identification of host powdery mildew fungi of the newly isolated *Ampelomyces* strains

Host powdery mildew fungi were identified based on morphology, host plants and rDNA-ITS sequences. Morphological observations of the powdery mildew hosts were done after preparing samples as described previously [58]. DNA extraction from powdery mildew mycelia was conducted as described [59]. For ITS amplifications in two steps [59], Phusion Green Hot Start II High-Fidelity PCR Master Mix (Thermo Fisher Scientific, MA, USA) was used with reaction mix composition and reaction conditions as recommended by the manufacturer and primer annealing temperature set to 58˚C. Tubes lacking DNA template were included as negative controls. Amplicons resulting from PCRs were run on 1% agarose gel. Fragments were sent for sequencing to LGC Genomics GmbH (Germany). Sequencing was done with the same primers used for the amplifications. The resulting chromatograms were processed using Staden Program Package [60]. Sequences were deposited to Genbank under accession numbers MT645844–MT645849.

### Isolation and culture of *Ampelomyces* strains

A total of 26 *Ampelomyces* strains were isolated from six powdery mildew fungal samples collected from their naturally infected host plants in three Prefectures in Japan (**Table 1**).

**Table 1. Designations, hosts, isolation data and GenBank accession numbers of the rDNA-ITS and *ACT* sequences of the 26 *Ampelomyces* strains included in the current work.**

| *Ampelomyces* strain designation[1] | Host powdery mildew species | Host plant species | Place of collection | Date of isolation | ITS accession numbers | *ACT* accession numbers |
|---|---|---|---|---|---|---|
| **Eq-h** | *Erysiphe quercicola* | *Quercus phillyraeoides* | Mie Prefecture, Japan | 12 Oct 2017 | MT644635 | MT656313 |
| Eq-k | | | | | MT644636 | MT656314 |
| **Eq-m** | | | | | MT644637 | MT656315 |
| **7100-a** | *Erysiphe glycines* | *Amphicarpaea bracteata* | Tochigi Prefecture, Japan | 18 Oct 2017 | MT644626 | MT656304 |
| **7100-b** | | | | | MT644627 | MT656305 |
| **7100-d** | | | | | MT644628 | MT656306 |
| 7100-e | | | | | MT644629 | MT656307 |
| **7100-g** | | | | | MT644630 | MT656308 |
| **7124-e** | *Erysiphe trifoliorum* | *Vicia* sp. | Shiga Prefecture, Japan | 02 Nov 2017 | MT644631 | MT656309 |
| 7134-b | *Erysiphe orixae* | *Orixa japonica* | Shiga Prefecture, Japan | 02 Nov 2017 | MT644632 | MT656310 |
| 7134-d | | | | | MT644633 | MT656311 |
| 7134-h | | | | | MT644634 | MT656312 |
| Xs-a | *Podosphaera xanthii* | *Xanthium stramonium* | Mie Prefecture, Japan | 06 Nov 2017 | MT644645 | MT656323 |
| Xs-c | | | | | MT644646 | MT656324 |
| Xs-h | | | | | MT644647 | MT656325 |
| Xs-m | | | | | MT644648 | MT656326 |
| Xs-s | | | | | MT644649 | MT656327 |
| **Xs-q** | | | | | MT644650 | MT656328 |
| Xs-y | | | | | MT644651 | MT656329 |
| Ga-a | *Golovinomyces asterum* var. *solidaginis* | *Solidago* sp. | Mie Prefecture, Japan | 05 Dec 2017 | MT644638 | MT656316 |
| Ga-b | | | | | MT644639 | MT656317 |
| Ga-c | | | | | MT644640 | MT656318 |
| Ga-d | | | | | MT644641 | MT656319 |
| Ga-e | | | | | MT644642 | MT656320 |
| Ga-f | | | | | MT644643 | MT656321 |
| Ga-g | | | | | MT644644 | MT656322 |

[1] Designations written in **bold** denote strains used in subsequent morphological and physiological analysis.

Pycnidia of *Ampelomyces* were isolated from powdery mildew fungi as described [25]. Colonies of *Ampelomyces* strains were cultured on Czapex-Dox agar medium supplemented with 2% malt extract (MCzA; 3g NaNO$_3$, 1g K$_2$HPO$_4$, 0.5g KCl, 0.5g MgSO$_4$, 15g agar and 20g malt extract) and maintained at 25 ± 2˚C and continuous illumination of 22.2 µmoL·m$^{-2}$·s$^{-1}$. Strains were subcultured every 2 months.

## DNA extraction from *Ampelomyces* strains, PCR amplifications and sequencing

DNA was extracted from approximately 1 cm$^2$ of two-week-old *Ampelomyces* colonies using DNeasy Plant Mini Kit (Qiagen, MD, USA) according to the protocol provided with the kit. Following DNA extraction, two loci, the rDNA-ITS and a fragment of the actin gene (*ACT*) were amplified and sequenced. For amplification of ITS, primers ITS1F [61] and ITS4 [62] and the following PCR protocol were used: initial denaturation at 94˚C for 5 min, followed by 35 cycles of 94˚C for 45 s, 55˚C for 45 s and 72˚C for 1 min. Reactions ended with final incubation at 72˚C for 10 min. Primers Act-1 and Act-5ra [63] were used for *ACT* amplifications with the

following PCR protocol: initial denaturation at 98˚C for 5 min, followed by 35 cycles of 98˚C for 30 s, 54˚C for 1 min and 72˚C for 1 min. Reactions ended with final incubation at 72˚C for 5 min. 20 μL reactions were run with DreamTaq Green PCR Master Mix (Thermo Fisher Scientific, MA, USA) using primers (Sigma-Aldrich, MO, USA) in 500 nM final concentration. PCR mixes lacking any DNA template were always included as negative controls. Sequencing and processing of chromatograms was conducted as detailed above. Sequences were deposited to Genbank under accession numbers MT644626–MT644651 (ITS) and MT656304–MT656329 (*ACT*).

## Sequence alignments and phylogenetic analyses

Sequences obtained from the new Japanese isolates were analyzed together with ITS and *ACT* sequences of 103 other *Ampelomyces* strains determined earlier [21, 26, 31, 55], and a *Phoma herbarum* strain (CBS 567.63) as an outgroup [21], thus the analysis included a total of 130 strains (including *Ph. herbarum*). Alignments and the final dataset to be analyzed were produced as described [55]. The combined ITS_*ACT* dataset was analyzed with maximum likelihood (ML) and Bayesian inference (BI) methods with partitions set corresponding to ITS and *ACT*. ITS consisted of 501 characters and *ACT* consisted of 820 characters, thus the combined dataset included 1321 characters in total. For ML analysis raxmlGUI 1.5 [64, 65] was used as detailed previously [55]. For BI, MrBayes 3.1.2 [66] was used and a previously described method [67] was followed. Majority (50%) rule consensus tree was calculated from the trees sampled in BI, omitting the first 4,000 sampled trees (burn-in). Phylogenetic trees resulting from analyses were visualized in TreeGraph 2.14.0 [68] and submitted to TreeBASE (study ID 25861).

## Morphological observations on *Ampelomyces in vitro* growth characteristics

Eight representative strains of *Ampelomyces*, originating from four powdery mildew samples, and belonging to four distinct haplotypes according to the phylogenetic analysis, were selected for a more detailed study of cultural characteristics (**Table 2**). The eight selected strains also included strains originating from the same powdery mildew sample, to reveal any potential strain-level differences in their characteristics. 30-day-old sporulating colonies of *Ampelomyces* strains were washed with 1.0–1.5 mL of sterile distilled water. Colonies were scraped

**Table 2. Morphological characteristics of *Ampelomyces* spp. fungi subcultured on Czapex-Dox agar medium supplemented with 2% malt (MCzA).**

| Strain designation | Spore length (μm) | | Spore width (μm) | | Germination rates (%) [a] | | Hyphal lengths (μm) [b] | | Colony areas (mm²) [c] | |
|---|---|---|---|---|---|---|---|---|---|---|
| Eq-h | 7.7 ± 1.5 | a | 4.0 ± 1.0 | a | 79.3 ± 9.0 | a | 44.5 ± 21.6 | a | 271.6 ± 119.7 | a |
| Eq-m | 6.6 ± 0.8 | a | 3.2 ± 0.5 | a | 74.2 ± 8.6 | ab | 29.7 ± 18.5 | b | 211.1 ± 46.3 | a |
| Xs-q | 7.0 ± 1.1 | a | 3.2 ± 0.6 | a | 80.0 ± 8.8 | a | 29.6 ± 18.1 | b | 263.1 ± 68.6 | a |
| 7100-a | 6.5 ± 0.8 | a | 3.0 ± 0.4 | a | 72.7 ± 8.0 | ab | 28.7 ± 16.9 | bc | 247.2 ± 79.8 | a |
| 7100-b | 6.6 ± 0.8 | a | 3.2 ± 0.5 | a | 81.5 ± 10.4 | a | 47.3 ± 21.9 | a | 274.8 ± 64.9 | a |
| 7100-d | 6.8 ± 0.7 | a | 3.3 ± 0.4 | a | 93.0 ± 2.2 | a | 55.6 ± 22.6 | d | 236.6 ± 88.1 | a |
| 7100-g | 7.3 ± 0.8 | a | 3.3 ± 0.5 | a | 69.1 ± 5.5 | b | 35.5 ± 21.1 | bc | 227.2 ± 118.2 | a |
| 7124-e | 7.2 ± 0.8 | a | 3.6 ± 0.4 | a | 94.2 ± 3.9 | a | 34.4 ± 14.5 | b | 228.7 ± 77.3 | a |

[a] Germination of *Ampelomyces* spores was observed 24 h after inoculation.

[b] Hyphal lengths of *Ampelomyces* strains were measured 48 h after inoculation.

[c] Colony areas of *Ampelomyces* strains were measured 20 days after inoculation of single pycnidia onto the center of MCzA medium.

Different letters in each column indicate significant difference ($p < 0.05$, Tukey's method).

with a sterile scalpel to produce spore suspensions from each colony. Their concentrations were measured with a haemocytometer (Nippon Rinsho Kikai Kogyo Co. Ltd., Tokyo, Japan). Suspensions were diluted to $5 \times 10^5$ spores·mL$^{-1}$. Tween 20 was added to reach a final Tween 20 concentration of 0.05%. Water agar plates overlaid with sterile cellophane sheets were sprayed each with the spore suspensions of one of the *Ampelomyces* strains and incubated at room temperature ($25 \pm 2°C$). Spore germination rates and hyphal lengths were measured 3 days after inoculation under a KH-2700 high-fidelity digital microscope (KH-2700 DM; Hirox, Tokyo, Japan). Data are presented as means and standard deviations of five replicates (100 spores at one replication). Radial growth rate of *Ampelomyces* spp. colonies was evaluated by measuring the diameter of each colony after incubation for 20 days. Data are presented as means and standard deviations of five replicates (10 colonies at one replication).

## Plant material

Seeds of the common tomato *Solanum lycopersicum* Mill. cv. Moneymaker (MM), obtained from self-pollinated progenies in our greenhouse, were germinated on water-soaked filter papers in a Petri dish for 3 days in an LH-240N growth chamber (Nippon Medical & Chemical Instruments, Osaka, Japan) under continuous illumination (19.8–40.3 µmoL·m$^{-2}$·s$^{-1}$; 400–700 nm) with white (full-spectrum) fluorescent lamps FL40SS W/37 (Mitsubishi, Tokyo, Japan) at $25 \pm 2°C$. Seedlings at the cotyledon stage were placed into polyurethane cubic sponge supports ($3 \times 3 \times 3$ cm$^3$). The sponge supports with seedlings were inserted into the 30 mL cylindrical plastic case (diameter 3 cm, length 5 cm) containing 20 mL hydroponic nutrient solution (4.0 mM $KNO_3$, 1.5 mM $Ca(NO_3)_2$, 1.0 mM $MgSO_4$, 0.66 mM $NH_4H_2PO_4$, 0.057 mM FeEDTA, 0.048 mM $H_3BO_3$ and 0.009 mM $MnSO_4$) and then incubated for 14 days in a temperature-controlled room under the following conditions: $25 \pm 2°C$, 50–70% relative humidity (RH) and continuous illumination of 59.5 µmoL·m$^{-2}$·s$^{-1}$.

## Powdery mildew material, inoculation, and incubation

A tomato powdery mildew isolate (*Pseudoidium neolycopersici* L. Kiss KTP-03, [69]), continuously maintained in the growth chamber was used in this study. Mature conidia were collected from the fungal colonies on infected leaves using an electrostatic spore collector as described previously [70] and transferred onto the true leaves of 14-day-old tomato healthy seedlings (MM) under a KH-2700 DM. The inoculated seedlings were incubated for 40 days in growth chambers at $25 \pm 1°C$ and 50–70% RH under continuous illumination of 22.2 µmoL·m$^{-2}$·s$^{-1}$ [69].

## Mycoparasitic tests with Japanese *Ampelomyces* strains

The eight *Ampelomyces* strains used in the morphological characterization study were included in the mycoparasitic tests. The following powdery mildew species maintained in the greenhouse were used: *Blumeria graminis* f. sp. *hordei* Marchal race 1 KBP-01 (on barley), *Erysiphe trifoliorum* Greville KRCP-4N (on red clover), *Podosphaera aphanis* U. Braun & S. Takamatsu KSP-7N (on strawberry), *Podosphaera xanthii* Pollacci KMP-6N (on melon) and *Pseudoidium neolycopersici* L. Kiss KTP-03 (on tomato). The powdery mildew colonies inoculated in these experiments were 10 days old. *Ampelomyces* spore suspensions were prepared as described above. Five powdery mildew infected plants for each strain were inoculated with spore suspensions of the tested *Ampelomyces* strain, and placed in plastic boxes. Five non-inoculated plants for each strain served as controls. Gauze pads were put in the boxes and wetted with sterile tap water. The boxes were closed and incubated for 10–20 days in growth chambers at $25 \pm 1°C$ and 80–90% RH under continuous illumination of 22.2 µmoL·m$^{-2}$·s$^{-1}$.

## Morphological analysis of pycnidia and conidia developed during mycoparasitism

*Ampelomyces* pycnidia and conidia developed during mycoparasitic tests were observed with an SZ60 stereomicroscope (SZ60 SM; Olympus, Tokyo, Japan) and a KH-2700 DM. Length and width of mature pycnidia and spores formed in tomato powdery mildew colonies grown for 14 days at 25 ± 2°C in the dark were measured on glass slides under a KH-2700 DM. Data are presented as means and standard deviations of five replicates (20 pycnidia and 100 spores at one replication).

## Quantification of mycoparasitic activity

To characterize the mycoparasitic activity of the *Ampelomyces* strains, the number of pycnidia formed in ten powdery mildew colonies of the five powdery mildew fungal species listed above was determined 14 days post inoculation (dpi). Number of pycnidia was scored on a three level scale of: no pycnidia present; from 11 to 100 pycnidia per powdery mildew colony; and 101 or more pycnidia formed on single powdery mildew colony. In addition, the number of *Ampelomyces* spores produced in each pycnidium developed in tomato powdery mildew was counted under a KH-2700 DM. Twenty leaf segments (approximately 1 x 1 cm$^2$ in size) were taken from ten powdery mildew-inoculated tomato plants. The samples were directly observed under the KH-2700 DM. Data are presented as means and standard deviations of five replicates (5 pycnidia at one replication).

## Consecutive monitoring of the mycoparasitic activities of *Ampelomyces* strains in the tomato powdery mildew fungus *Ps. neolycopersici*

Strain Xs-q, one of the strains showing reliable and intensive *in vitro* sporulation, was selected for a more detailed analysis on the mycoparasitic activity inside powdery mildew structures. To gain a general overview on the process of mycoparasitism and to observe the development of pycnidia in detail, Xs-q was spray-inoculated onto powdery-mildew infected tomato plants as detailed above. Inoculated leaves were observed regularly using the KH-2700 DM. In addition, samples were prepared for a BX-60 light microscope (BX-60 LM; Olympus, Tokyo, Japan). For this, samples were fixed and chlorophyll was removed in a boiling alcoholic lactophenol solution (10 mL glycerol, 10 mL phenol, 10 mL lactic acid, 10 mL distilled water and 40 mL 99.8% ethanol) for 1–2 min and then stained with 0.1% Aniline Blue (Nacalai Tesque, Tokyo, Japan) dissolved in distilled water as described previously [71]. The samples were observed under a BX-60 LM.

To reveal more details of mycoparasitism, and in particular to facilitate the direct observation of the infection process of *Ampelomyces* strains real-time, the following experiment was designed. Mature conidia of *Ps. neolycopersici* KTP-03 were inoculated directly onto type I trichomes of 14-day-old MM plants [70] and the inoculated plants were incubated in the growth chamber for 10–14 days under the same conditions as described above. Powdery mildew grown on trichomes provided natural environmental conditions which are also well suited for observations [57]. The infection process of *Ampelomyces* strains in KTP-03 was observed using the KH-2700 DM. For this, single mature pycnidia of the *Ampelomyces* strain Xs-q grown on MCzA plates were placed onto *Ps. neolycopersici* KTP-03 colonies using glass needle micromanipulator [72]. These single pycnidia served as a source of conidia, initiating the infection process to be observed. Hyphal development of the strain Xs-q was consecutively photographed during a 14-day period after inoculation, using the 1/2″ Interline transfer charge-coupled device (CCD) camera of the KH-2700 DM. The micrographs were analyzed

using Adobe Photoshop image-processing software (ver. 5.0; Adobe Systems, CA, USA) for better contrasting of the captured images without changing the original information.

## Results

### Sequence analysis of ITS and *ACT* regions of *Ampelomyces* isolates

ITS and a fragment of *ACT* were determined in a total of 26 *Ampelomyces* strains newly isolated in Japan (**Table 1**). ITS and *ACT* sequences of *Ampelomyces* strains isolated from the same powdery mildew samples were identical. Strains from the six independent powdery mildew samples represented five distinct ITS genotypes, as strains isolated from *Erysiphe quercicola* and *E. glycines* had identical ITS sequences. *ACT* sequences revealed six distinct genotypes.

### Analysis of phylogeny of *Ampelomyces* isolates

ML and BI analyses on the combined ITS_*ACT* dataset revealed very similar phylogenies and identical groupings of clades. The tree with the highest likelihood is shown in **Fig 1**. Phylogenetic analysis of *Ampelomyces* strains revealed five distinct, well supported clades (bootstrap supports $\geq$ 94 percent; posterior probabilities = 1). The largest clade, designated as clade 1, consisted of 60 strains, including eight *Ampelomyces* strains isolated in this work. This clade included several subgroups. The eight Japanese strains clustered in clade 1 belonged to two distinct subgroups. Clade 2 contained 13, and clade 3 contained 15 strains, the latter including seven newly isolated strains. Clade 4, the second-largest clade was composed of 22 strains, half of which were isolated in this work. Thus, most of the strains isolated in this work belonged to this clade, incorporating three different genotypes of Japanese strains. Lastly, clade 5 contained 19 strains, none of which was isolated in Japan. Consequently, strains isolated from six powdery mildew samples collected in Japan were accommodated in three major distinct clades of *Ampelomyces*.

### Morphological observation of Japanese *Ampelomyces* strains grown *in vitro*

Morphological characteristics were observed in detail in eight selected isolates. The following characteristics were determined: spore sizes and shapes, spore germination rates, time needed for germination, hyphal lengths, and colony areas. The spores with unicellular, hyaline and ellipsoid-ovoid to doliiform, *ca*. 5.7–9.2 x 2.6–5.0 μm. Spores germinated *ca*. 15–20 h after inoculation, elongated hyphae and then branched under conditions of high RH. The lengths of the hyphae formed from spores were *ca*. 6.2–78.2 μm 48 h after inoculation. Fungal colonies slowly and concentrically spread after inoculation of a single mature pycnidium on the center of MCzA media. The colony areas were *ca*. 148.4–391.3 mm$^2$ 20 dpi. There were significant differences in germination rates and hyphal lengths among the isolates, but not in colony areas (**Table 2**).

### Mycoparasitic tests with Japanese *Ampelomyces* strains

Eight Japanese *Ampelomyces* strains were individually inoculated onto five different powdery mildew fungi maintained in our laboratory. All strains successfully infected all the five powdery mildew isolates and formed mature pycnidia in four out of five mycohost colonies (*E. trifoliorum*, *Po. aphanis*, *Po. xanthii* and *Ps. neolycopersici*), but not in *B. graminis* on barley. The tested strains infected the melon powdery mildew KMP-6N more heavily than others, as reflected by forming more pycnidia than in the other tested powdery mildew isolates 14 dpi.

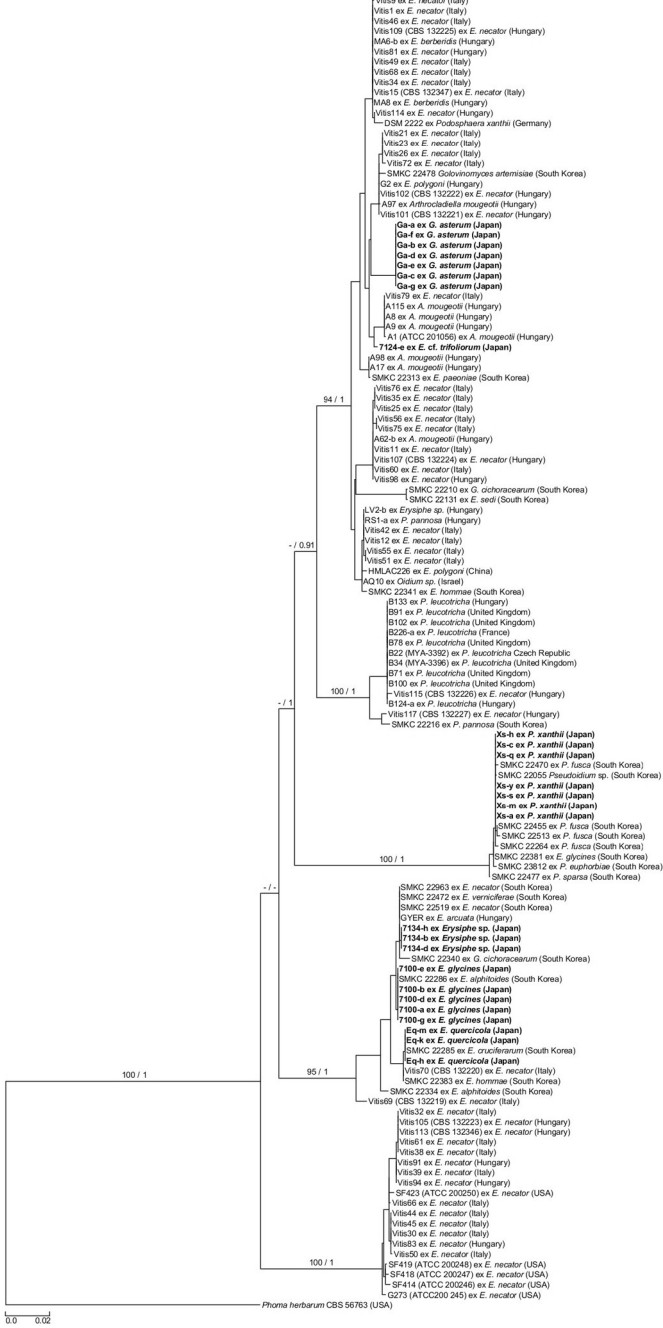

**Fig 1. Phylogenetic tree with the highest likelihood value resulting from the maximum likelihood (ML) analysis of ITS and *ACT* sequences of 129 *Ampelomyces* strains, including 26 isolated in this study (written in bold) in Japan.** The tree is rooted to *Phoma herbarum* strain CBS56763 based on an earlier study [17]. Bootstrap values calculated from 1000 replicates in ML analysis are written as percentages (values below 70% and in subclades are not shown). These are followed by posterior probabilities resulting from Bayesian analysis (no values are shown below 0.9 and in subclades), separated by slashes. Bar indicates 0.02 expected changes per site per branch. Additional data on *Ampelomyces* strains (place and data of collection, host fungal species) and GenBank accession numbers of sequences included in the analyses are listed in earlier studies [17, 22].

There were no significant differences in the mycoparasitic activity of the eight *Ampelomyces* strains based on the three-level scoring (**Table 3**).

**Table 3. Results of the mycoparasitic tests with eight *Ampelomyces* strains and five powdery mildew species, and morphological characteristics of pycnidia produced in conidiophores of tomato powdery mildew KTP-03.**

| Strains | Formation of mature pycnidia in/on hyphae of powdery mildew fungi | | | | | Pycnidia of *Ampelomyces* in tomato powdery mildew conidiophores | | | | Number of spores produced in mature pycnidia in tomato powdery mildew conidiophores [c] | |
|---|---|---|---|---|---|---|---|---|---|---|---|
| | Barley PM [a] | Melon PM | Red clover PM | Straw-berry PM | Tomato PM | Length (μm) | | Width (μm) | | | |
| Eq-h | – [b] | ++ | + | + | + | 72.4 ± 11.8 | a | 40.8 ± 7.3 | a | 426.3 ± 222.8 | bc |
| Eq-m | – | ++ | + | + | + | 57.0 ± 15.6 | ab | 31.0 ± 7.7 | ab | 552.8 ± 206.1 | b |
| Xs-q | – | ++ | + | + | + | 55.1 ± 14.8 | b | 38.9 ± 7.6 | a | 493.2 ± 112.6 | b |
| 7100-a | – | ++ | + | + | + | 63.7 ± 11.3 | a | 35.3 ± 6.0 | a | 1064.9 ± 427.8 | a |
| 7100-b | – | ++ | + | + | + | 59.6 ± 16.3 | ab | 32.7 ± 6.4 | ab | 1025.0 ± 274.2 | a |
| 7100-d | – | ++ | + | + | + | 67.0 ± 11.5 | a | 36.5 ± 6.2 | a | 590.8 ± 183.3 | bc |
| 7100-g | – | ++ | + | + | + | 62.0 ± 16.6 | ab | 29.3 ± 6.7 | b | 317.2 ± 117.8 | c |
| 7124-e | – | ++ | + | + | + | 53.6 ± 13.4 | b | 30.8 ± 7.6 | ab | 569.0 ± 255.7 | b |

[a] 10-day-old powdery mildew colonies were used in this experiment.

[b] To characterize the mycoparasitic activity of the *Ampelomyces* strains in five powdery mildew species, the numbers of pycnidia formed inside single powdery mildew colonies were counted 14 dpi: –, +, and ++ represent no pycnidia, from 11 to 100 pycnidia, and more than 101 pycnidia, respectively.

[c] The number of spores in each newly formed pycnidium was determined 14 dpi.

Different letters in each column indicate significant difference ($p < 0.05$, Tukey's method).

## Morphology of pycnidia developed in tomato powdery mildew colonies and quantification of mycoparasitism

Sizes and shapes of the newly developed pycnidia and the number of spores produced in a pycnidium in tomato powdery mildew colonies were observed in detail and data are shown in **Table 3**. Pycnidia started to form *ca*. 6–7 dpi; the inoculated KTP-03 mycelia were 10-day-old at the time of the spray inoculations. **Fig 2A** shows the mature, black coloured and ovoid pycnidia (Py1), and smaller immature pycnidia with pale yellow colour (Py2). Pycnidia were mainly produced in conidiophores of mycohosts (**Fig 2B**). Mature pycnidia measured *ca*. 40.2–84.2 x 22.6–48.1 μm, and the numbers of spores produced in a single mature pycnidium were *ca*. 199.4–1492.7 14 dpi. As shown in **Fig 2C**, numerous spores were released from intracellular pycnidia by the rupture of the pycnidial wall after adding a 10 μL drop of distilled water. There were significant differences in the number of spores developed in a single pycnidium, and also the sizes of pycnidia among the strains (**Table 3**).

## Infection process of *Ampelomyces* strain Xs-q in *Ps. neolycopersici*

*Ps. neolycopersici* KTP-03 formed lobed primary appressoria, elongated colony-forming hyphae from the conidial bodies within 48 h after inoculation of conidia onto the type I trichome cells, and successfully infected the trichomes. KTP-03 hyphae growing from the trichomes developed vigorously on the epidermal cells. KTP-03 formed conidiophores within 7 days after attachment of the tip of hyphae from the trichome cells on the epidermal cells (**Fig 3A**). After spray inoculation of Xs-q spores, the spores germinated on tomato leaves and penetrated the KTP-03 conidia (**Fig 3B**) or/and hyphae, and continued their growth internally from cell to cell of KTP-03 hyphae (**Fig 3C**). Eventually, Xs-q produced new intracellular pycnidia in KTP-03 conidiophores *ca*. 8–10 dpi (**Fig 3D**) and released spores from mature pycnidia *ca*. 10–14 dpi (**Fig 3E**). The mature spores, which were released from intracellular pycnidia by the rupture of the pycnidial cell walls, became the sources of subsequent infections of the powdery mildew host. Finally, KTP-03 hyphae were destroyed by invasion of the mycoparasite (**Fig 3F**).

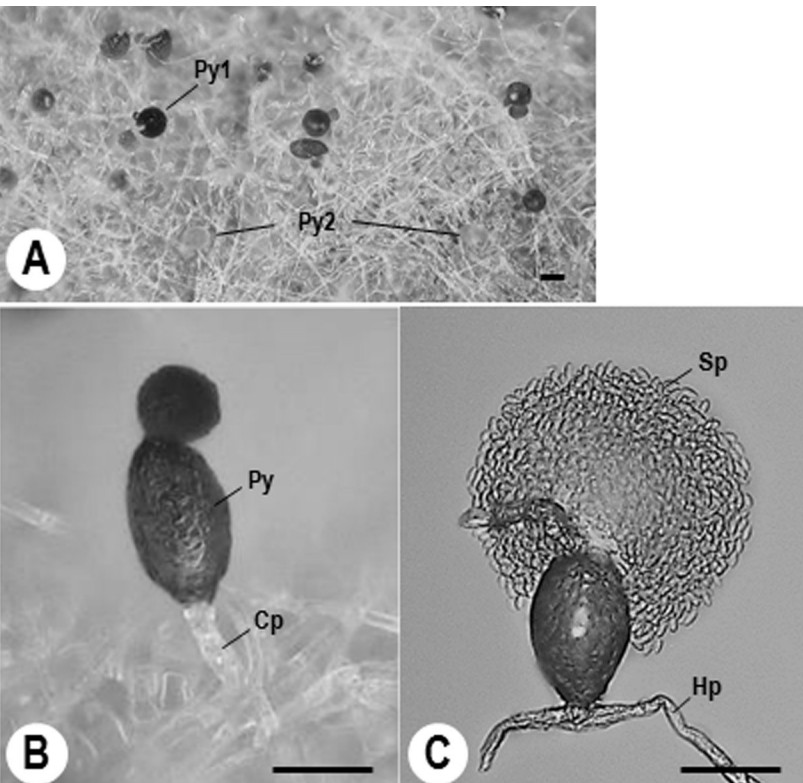

**Fig 2.** Digital (A and B) and light micrographs (C) of *Ampelomyces* strain Xs-q on tomato powdery mildew KTP-03 mycelia. A, mature (Py1) and immature pycnidia (Py2) formed onto KTP-03 mycelia. B, Pycnidium (Py) produced in KTP-03 conidiophores (Cp). C, Spores (Sp) appeared from a pycnidium after adding a drop of distilled water (10 μL). Micrographs were taken at 14 days after spray inoculation of spores onto 10-day-old KTP-03 mycelia. Bars represent 50 μm (A) and 30 μm (B and C). Hp, Hypha of tomato powdery mildew KTP-03.

During mycoparasitism, we focused on the invasion into host structures, and the development of intracellular pycnidia in particular. We consecutively observed the invasion process of strain Xs-q in KTP-03 hyphae using a KH-2700 DM, taking advantage of KTP-03-infected type I trichome cells. After inoculation of single Xs-q pycnidia onto tomato leaf in close vicinity of the trichomes (**Fig 4A**), the Xs-q spores released from the pycnidia germinated, developed hyphae which elongated towards KTP-03 hyphae (**Fig 4B and 4C**). By invasion of Xs-q hyphae into KTP-03 hyphae, KTP-03 conidiophores and hyphae initiated atrophy 5 dpi (**Fig 4D**); atrophied 6 dpi (**Fig 4E**); then completely collapsed 7 dpi (**Fig 4F**). The intracellular Xs-q hyphae grew out from conidia formed at top of the conidiophores (**Fig 4D–4F**).

For the detailed observations on pycnidial development, tomato powdery mildew colonies infected with *Ampelomyces* strains following spray inoculation were used. **Fig 5A** shows well developed KTP-03 conidiophores. Foot and generative cells of KTP-03 conidiophores showed the first signs of atrophy *ca*. 5–6 dpi (**Fig 5B**). Intracellular pycnidia of Xs-q started to be produced mostly in the basal cells of the conidiophores *ca*. 6–8 dpi (**Fig 5C**). In the meantime, Xs-q hyphae and pycnidia continued to elongate in KTP-03 hyphae (**Fig 5D**). Immature Xs-q pycnidia were pale yellow, and then became dark brown or almost black after maturation. Consequently, the conidiophores completely collapsed *ca*. 10–14 dpi (**Fig 5E**). High numbers of spores were released from the mature pycnidia after adding 20 μL drops of distilled water to the parasitized colonies. The released mature spores served as sources for subsequent infections to KTP-03 hyphae.

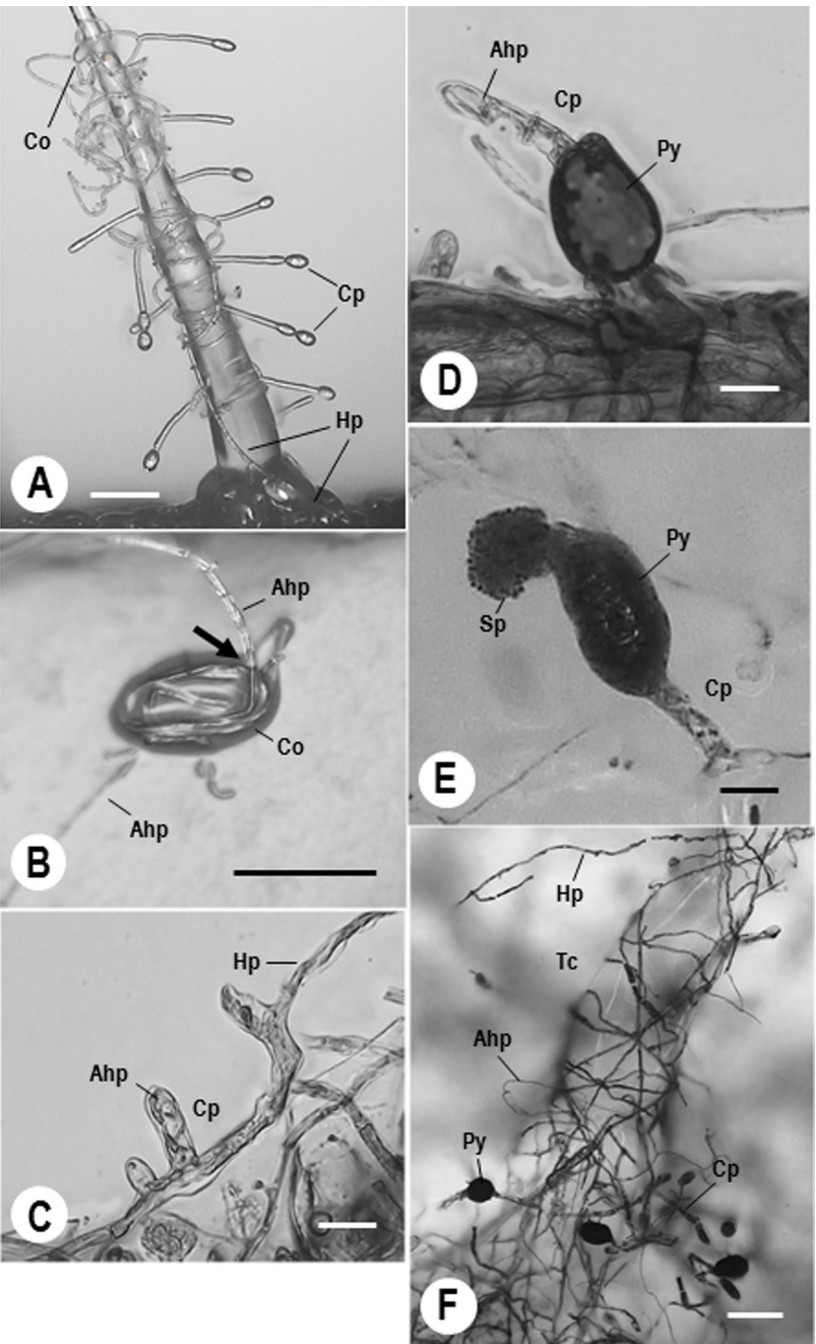

**Fig 3.** Digital (A and B) and light micrographs (C to F) of *Ampelomyces* strain Xs-q in tomato powdery mildew KTP-03. A, KTP-03 hyphae infected tomato type I trichome cells. B, Xs-q hyphae invaded into KTP-03 conidium. Arrow shows invasion site of Xs-q hypha. C, Xs-q hyphae grown into KTP-03 hyphae and primary conidiophore. D, Xs-q hyphae grown and pycnidium produced in a KTP-03 conidiophore. E, Xs-q pycnidium produced in a KTP-03 conidiophore and abundant spores appeared from the pycnidium. F, Xs-q pycnidia produced in KTP-03 conidiophores formed on tomato type I trichome cell. Bars represent 60 μm (A and F) and 20 μm (B to E). Ahp, Xs-q hypha; Co, KTP-03 conidium; Cp, KTP-03 conidiophore; Hp, KTP-03 hypha; Py, Xs-q pycnidium; Sp, Xs-q spore; Tc, Tomato type I trichome cell.

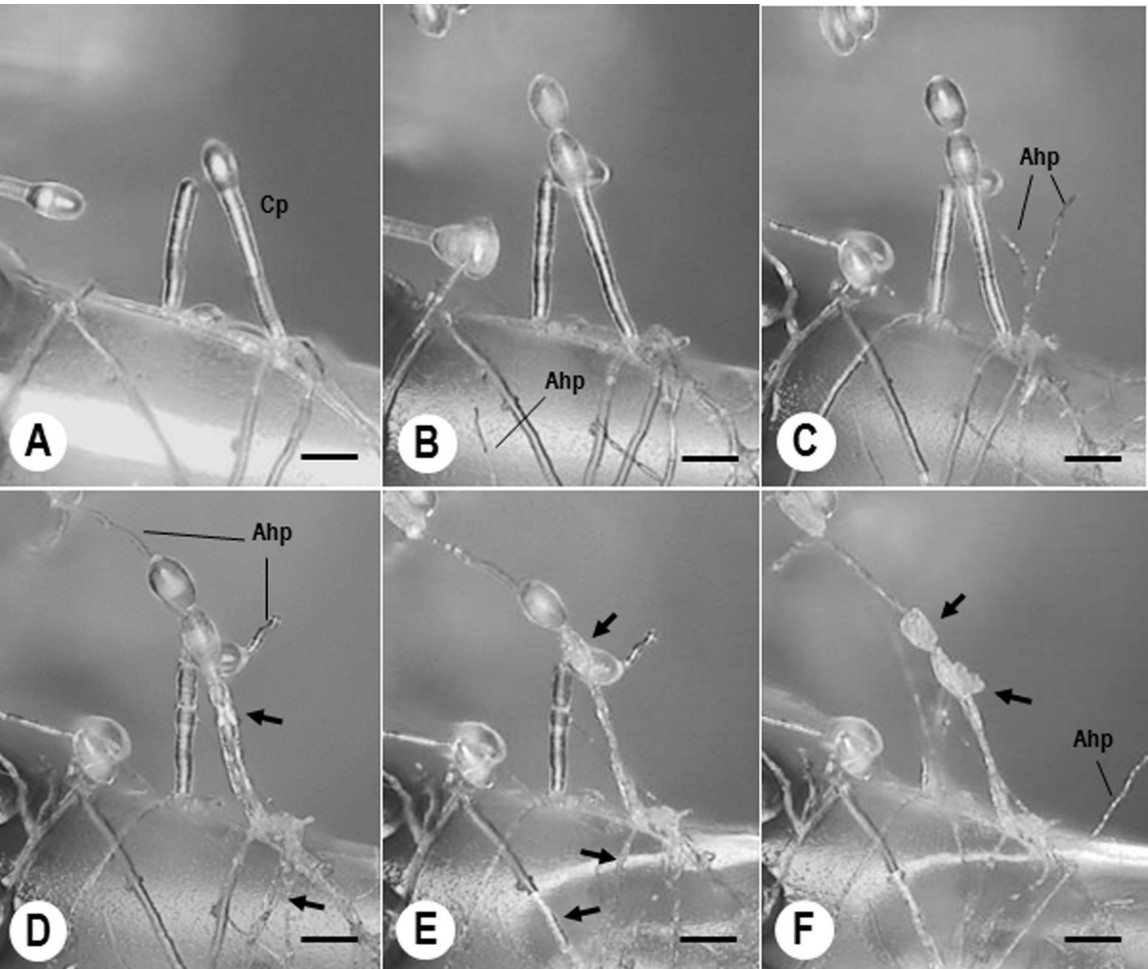

**Fig 4. Infection process of *Ampelomyces* strain Xs-q in tomato powdery mildew KTP-03.** KTP-03 vigorously grew on tomato type I trichome cells. Digital micrographs were taken at 0 (A), 2 (B), 3 (C), 5 (D), 6 (E) and 7 days (F) after inoculation of a pycnidium onto tomato leaf in the vicinity of the trichome cells. KTP-03 conidiophores (Cp) and hyphae (arrowed) atrophied after invasion of Xs-q hyphae (Ahp) into KTP-03 hyphae. Bars represent 20 μm.

## Discussion

In this study, we isolated a total of 26 *Ampelomyces* strains, potential BCAs of crop powdery mildews [20], from six powdery mildew species in Japan, and characterized the strains based on morphology and molecular phylogenetic analyses. To the best of our knowledge, this is the first detailed analysis of *Ampelomyces* strains isolated in Japan. The morphological characteristics of Japanese strains were similar to those described by other works [73, 74]. Our isolates always grew slowly in culture, with an *in vitro* radial growth rate of 0.5–1.0 mm·d$^{-1}$ on MCzA, close to the fungal development rates reported for other *Ampelomyces* isolates [45, 75]. Based on morphological analysis and *in vitro* growth rates we concluded that our isolates represented true *Ampelomyces*, intracellular mycoparasites of powdery mildew fungi.

Molecular analyses based on ITS sequences, and also *ACT* sequences, have revealed considerable genetic diversity among *Ampelomyces* strains [21, 23, 25, 26, 29, 55]. Using sequences from these two loci, we confirmed the existence of at least five different phylogenetic lineages within the genus *Ampelomyces* reported earlier, and showed that the newly isolated Japanese strains belong to three major clades. We analyzed phenotypic characteristics of *Ampelomyces*

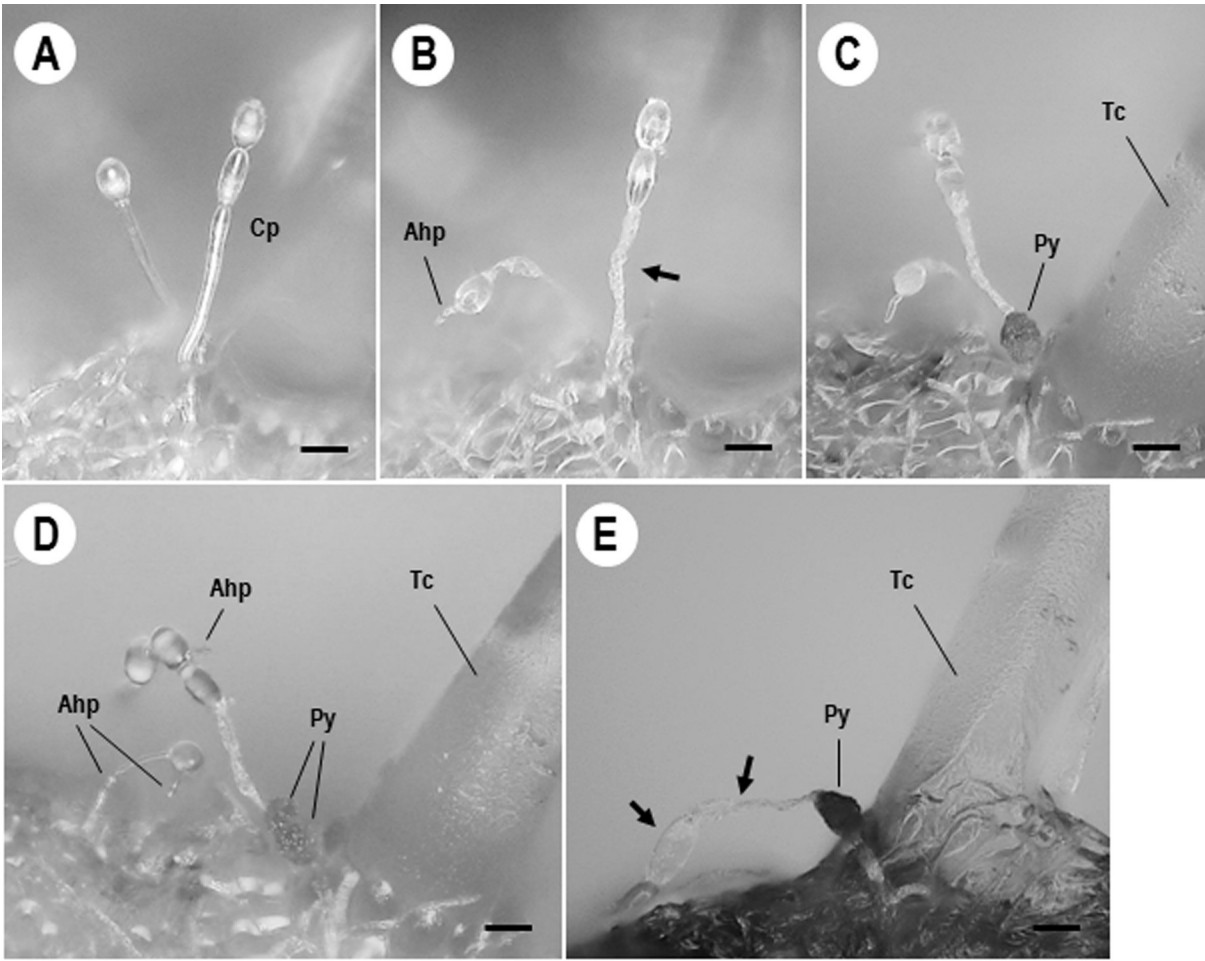

**Fig 5. Pycnidial formation processes of *Ampelomyces* strain Xs-q in KTP-03.** A KTP-03 mycelium was formed on tomato leaf at 10 days after inoculation of a KTP-03 conidium with micromanipulation technique. Digital micrographs were taken at 0 (A), 5 (B), 7 (C), 10 (D) and 12 days (E) after spray inoculation of Xs-q spores onto the 10-day-old KTP-03 mycelium. KTP-03 conidiophores (Cp), formed near the trichome cell (Tc), completely atrophied after invasion of Xs-q hyphae (Ahp) into KTP-03 hyphae (arrows). Xs-q pycnidia (Py) were successfully produced in KTP-03 conidiophores. Bars represent 20 μm.

strains isolated from four different powdery mildew samples, and also four different strains isolated from the same powdery mildew sample. In our results, we did not detect morphological characteristics which could clearly be associated with given genotypes or clades. Four strains isolated from the same powdery mildew sample, however, displayed significant differences in measured hyphal lengths, germination rates and number of spores developed in single pycnidia. These undoubtedly show the presence of differences on strain level, a phenomenon also observed in other works [27, 30]. It is currently unknown if the differences in the phenotypic characteristics of different strains of *Ampelomyces* are somehow related to their yet unrevealed genetic diversity or are simply caused by phenotypic plasticity. The possibility of strain-level differences, however, needs to be taken in consideration in upcoming studies, such as during development of future biocontrol products.

We conducted tests using spore suspensions of *Ampelomyces* strains. The concentration of spore suspensions is an important factor affecting their germination and infection. Germination of *Ampelomyces* spores decreases rapidly when the spores are at a concentration of more than $10^6$ spores·mL$^{-1}$, due to the production of self-inhibitory substances [76]. Therefore, in

this study, we used a lower concentration of spores ($5 \times 10^5$ spores·mL$^{-1}$) for spray inoculation onto nutrient media and powdery mildew-infected leaves. Japanese strains of *Ampelomyces* successfully germinated at 10–20 h after spray inoculation of the spores onto MCzA and powdery mildew-infected plants under conditions of high RH.

Earlier experiments have shown that *Ampelomyces* mycoparasites collected from a given powdery mildew species can produce intracellular pycnidia in mycelia of other species of the Erysiphaceae [28, 77, 78]. After the inoculation tests with eight Japanese isolates and five powdery mildew species, we observed the degeneration and constriction of parasitized hyphae of all five powdery mildew species tested, and also pycnidial formation in the hyphae and conidiophores of four powdery mildew fungi. These results showed that the Japanese *Ampelomyces* strains infected powdery mildew hyphae irrespective of the original host and produced intracellular pycnidia in mycelia of four out of five tested mycohosts. This is in accordance with several other studies [29–31, 43, 77] reporting lack of host specificity with the tested *Ampelomyces* strains. However, a previous study reported the lack of pycnidial production of a strain isolated from *E. artemisiae* in *B. graminis* on barley [73] similar to what was observed in this study. Other studies showed typical mycoparasitism, including formation of intracellular pycnidia, in *B. graminis* conidiophores on cereals (wheat and barley) by *Ampelomyces* strains isolated from powdery mildews infecting dicots [28, 33, 46, 55]. Thus, previous mycoparasitic tests with *B. graminis* infecting cereals led to contradictory results. These results, together with the weak mycoparasitism of *B. graminis* observed in our experiments might have been due to unfavorable experimental conditions [20].

In an earlier study [57], it was demonstrated that the tomato trichomes provide an interaction site that can be used as infection sites of powdery mildew fungi. We took advantage of trichome cells projecting from host plant leaves and extended the use of the method for the detailed morphological analysis of mycoparasite-mycohost interactions, determining how and when the mycoparasites invade powdery mildew structures. Based on earlier works and on our results obtained with detailed microscopic analysis, the approximate time course of the infection, the developmental process of the infection of the mycoparasites in the mycohosts and morphological changes of mycoparasite-infected powdery mildew fungi can be summarized as follows. *Ampelomyces* hyphae are spread by wind within the parasitized powdery mildew conidia [38, 79, 80] and spores are dispersed by rain splash [81, 82]. When the mycohost is present, *Ampelomyces* is able to penetrate and parasitize hyphae of powdery mildew fungi through mechanical [81] or enzymatic processes [30, 83] after germination. The penetration into mycohost structures can occur within 24 h [42, 81]. Then, the hyphae of the mycoparasite continue their growth in powdery mildew structures, from cell to cell through the septal pores [81, 84]. The invasion of *Ampelomyces* into the mycohost leads to atrophy in 5–6 days and then to complete disruption of the mycohost conidiophores 7 dpi. Disruption of the cytoplasm of the fungal hosts will result in reduced growth, and eventually death of the host fungus [18, 43, 84]. During the course of the infection, *Ampelomyces* produces intracellular pycnidia in the hyphae or conidiophores of mycohosts 5 to 10 dpi [36, 78, 81]. Based on our digital microscopic observation, we showed visually that pycnidial production started 6–8 dpi, and pycnidia were morphologically mature 8–10 dpi. Also, we confirmed that almost all pycnidia were produced in conidiophores of the mycohost (see **Fig 3**). Eventually, we clarified that spores were released from intracellular pycnidia by the rupture of both the pycnidial and the powdery mildew cell walls, which in our experiments happened *ca.* 10–14 dpi. Understanding of the infection mechanisms of the *Ampelomyces*, including timing and quantifying mycoparasitism of *Ps. neolycopersici* on tomato (e.g. type I trichomes), will be useful for analyzing specific gene expression regarding the invasion and infection of mycoparasite fungi into powdery mildew hyphae during various infection stages of *Ampelomyces* strains in the future study.

To our knowledge, this is the first study that has documented the developmental process of *Ampelomyces* mycoparasites in a powdery mildew fungus using light and digital microscopes, revealing the time course of the mycoparasitic interaction and providing quantitative data to characterize the sporulation rate of *Ampelomyces* strains inside powdery mildew conidio-phores. The developed model system will be used in future studies to analyze expression levels of specific genes involved in mycoparasitism, and also to improve practical strategies to control powdery mildew infections of crops. For example, *Ampelomyces* has been reported to tolerate a number of fungicides, acaricides and insecticides applied in plant protection [38, 39, 42, 78]. As a next step, we will examine the tolerance of Japanese *Ampelomyces* strains to commercial pesticides and focus on developing reliable BCAs against economically important crop pow-dery mildews.

## Author Contributions

**Conceptualization:** Levente Kiss, Teruo Nonomura.

**Data curation:** Márk Z. Németh, Yuusaku Mizuno, Levente Kiss, Teruo Nonomura.

**Formal analysis:** Márk Z. Németh, Yuusaku Mizuno, Susumu Takamatsu, Yoshinori Matsuda.

**Funding acquisition:** Márk Z. Németh, Susumu Takamatsu, Teruo Nonomura.

**Investigation:** Márk Z. Németh, Yuusaku Mizuno, Hiroki Kobayashi, Diána Seress, Naruki Shishido, Yutaka Kimura, Tomoko Suzuki, Yoshihiro Takikawa, Koji Kakutani.

**Methodology:** Márk Z. Németh, Yuusaku Mizuno, Diána Seress, Yoshinori Matsuda.

**Project administration:** Márk Z. Németh, Levente Kiss, Teruo Nonomura.

**Resources:** Márk Z. Németh, Susumu Takamatsu, Levente Kiss, Teruo Nonomura.

**Supervision:** Susumu Takamatsu, Levente Kiss, Teruo Nonomura.

**Visualization:** Márk Z. Németh, Levente Kiss, Teruo Nonomura.

**Writing – original draft:** Márk Z. Németh, Yuusaku Mizuno.

**Writing – review & editing:** Levente Kiss, Teruo Nonomura.

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
