## [Decision Letter · Decision Letter 0]

12 Apr 2021

PONE-D-20-39222

Ampelomyces strains isolated from diverse powdery mildew hosts in Japan: Their phylogeny and mycoparasitic activity, including timing and quantifying mycoparasitism of Pseudoidium neolycopersici on tomato

PLOS ONE

Dear Dr. Teruo Nonomura,

Thank you for submitting your manuscript to PLOS ONE. After careful consideration, we feel that it has merit but does not fully meet PLOS ONE’s publication criteria as it currently stands. Therefore, we invite you to submit a revised version of the manuscript that addresses the points raised during the review process.

We look forward to receiving your revised manuscript.

Kind regards,

Tofazzal Islam, Ph.D.

Academic Editor

PLOS ONE

Journal Requirements:

4. We suggest you thoroughly copyedit your manuscript for language usage, spelling, and grammar. If you do not know anyone who can help you do this, you may wish to consider employing a professional scientific editing service.  

6. Thank you for stating the following in the Funding Section of your manuscript:

[This work was partly supported by Grants for Scientific Research from Faculty of Agriculture, Kindai University, and Agricultural Technology and Innovation Research Institute, Kindai University. MZ Németh’s stay in Japan was supported by the Institute of Fermentation, Osaka.The funders had no role in study design, data collection and analysis, decision to publish, or preparation of the manuscript.]

 [No. The funders had no role in study design, data collection and analysis, decision to publish, or preparation of the manuscript.]

Reviewers' comments:

Reviewer's Responses to Questions

**Comments to the Author**

1. Is the manuscript technically sound, and do the data support the conclusions?

Reviewer #1: Yes

Reviewer #2: Yes

Reviewer #3: Yes

2. Has the statistical analysis been performed appropriately and rigorously? 

Reviewer #1: Yes

Reviewer #2: Yes

Reviewer #3: Yes

3. Have the authors made all data underlying the findings in their manuscript fully available?

Reviewer #1: Yes

Reviewer #2: Yes

Reviewer #3: Yes

4. Is the manuscript presented in an intelligible fashion and written in standard English?

Reviewer #1: Yes

Reviewer #2: Yes

Reviewer #3: Yes

5. Review Comments to the Author

Reviewer #1: This is a strong manuscript that presents an analyses of an important mycoparasite that may have agricultural use. There are only minor comments in the manuscript. I do ask for comment on Epicoccum. This is mentioned in the discussion as explaining divergence in one of the clades. Epicoccum is in a different family. It is often isolated as an endophyte suggesting that perhaps sequences used where somehow misidentified. This should be resolved or expanded.

Reviewer #2: Straight forward manuscript on a very interesting group of fungi. I have only minor revisions, mostly related to improvements for the discussion. Authors should focus on making the discussion more concise. Don't just repeat results, rather discuss how results improves understanding of Ampelomyces. Focus on the "including timing and quantifying mycoparasitism of Pseudoidium neolycopersici on tomato" portion of your title.

In the abstract, why not end with relevance in regards to developing a model system to further examine the use of Ampelomyces for biocontrol? Just emphasizing the strains origins from Japan or the use of digital microscopy is not quite novel, compared to the description of a model system to further study Ampelomyces mycoparasitism.

Reviewer #3: This is a very interesting research. The phylogeny and mycoparasitic activity of Ampelomyces strains isolated from diverse powdery mildew hosts in Japan were explored. I recommend to accepting it for publishing. A few comments are shown in trace in the attached pdf file.

6. PLOS authors have the option to publish the peer review history of their article (what does this mean?). If published, this will include your full peer review and any attached files.

Reviewer #1: No

Reviewer #2: No

Reviewer #3: **Yes: **Shu-Yan LIU

---

## [Author Response · Author response to Decision Letter 0]

23 Apr 2021

Dear Dr. Tofazzal Islam 

PLOS ONE

We would like to express our sincere thanks for the editor’s and reviewers’ large efforts to improve our manuscript (Ms. Ref. No PONE-D-20-39222). We attempted to faithfully respond to the editor’s and reviewers’ comments given to us. According to the editor’s and reviewers’ request, we indicated changes with line numbers corresponding to line numbering of the revised manuscript (lines numbers of the final text, without showing track changes).

With best regards, 

Dr. Teruo Nonomura

Kindai University, Japan

Dr. Levente Kiss

University of Southern Queensland, Australia

Journal Requirements

►Response to 1

Style was altered to meet PLOS ONE's style requirements.

2. Please review your reference list to ensure that it is complete and correct. 

►Response to 2

Reference list was reviewed and it is considered complete and correct.

3. PLOS requires an ORCID iD for the corresponding author.

►Response to 3

Corresponding authors ORCID iD was added.

Prof. Teruo Nonomura ORCID ID: 0000-0002-0249-3369

Prof. Levente Kiss ORCID ID: 0000-0002-4785-4308

4. We suggest you thoroughly copyedit your manuscript for language usage, spelling, and grammar. 

►Response to 4

Done.

5. We note that you have included the phrase “data not shown” in your manuscript. Unfortunately, this does not meet our data sharing requirements […] if the data are not a core part of the research being presented in your study, we ask that you remove the phrase that refers to these data. 

►Response to 5

We have amended the sentence and removed the phrase “data not shown”. This does not cause any changes in the results and their interpretation.

6. We note that you have provided funding information that is not currently declared in your Funding Statement. However, funding information should not appear in the Acknowledgments section or other areas of your manuscript. Please remove any funding-related text from the manuscript and let us know how you would like to update your Funding Statement. Please include your amended statements within your cover letter; we will change the online submission form on your behalf. 

►Response to 6

We indicated our intention to amend the Funding Statement in the cover letter.

Reviewer's Comments to the Authors

Reviewer #1: This is a strong manuscript that presents an analyses of an important mycoparasite that may have agricultural use. There are only minor comments in the manuscript. I do ask for comment on Epicoccum. This is mentioned in the discussion as explaining divergence in one of the clades. Epicoccum is in a different family. It is often isolated as an endophyte suggesting that perhaps sequences used where somehow misidentified. This should be resolved or expanded.

►Response to Reviewer #1

We thank Reviewer #1 for the positive assessment of our manuscript, for the review and for the comments and corrections.

Minor comments were addressed in the revised manuscript (rephrasing, typo corrections) at all the positions marked by the Reviewer in the PDF file.

About the comment on Epicoccum: Thank you for pointing this out. Earlier studies on pycnidial fungi isolated from powdery mildew colonies treated all those fungi as 'Ampelomyces', based mostly on pycnidial morphology. Later, molecular phylogenetic studies revealed that not all those pycnidial fungi isolated from powdery mildew colonies belong to the genus Ampelomyces. As suggested by the Reviewer, we tried to amend the corresponding part (P21 L417-421) of the ‘Discussion’ section in revised manuscript (RM). Here we decided to avoid directly mentioning Epicoccum, in order to maintain what we intended, but also to make the paragraph easy to understand.

We hope that the corrections based on the Reviewer’s suggestions will overcome the concerns described in the review.

Additional comments:

* rephrase here "this is how"

►Response

We deleted the sentence (P24 L476-477 in original manuscript) in the ‘Discussion’ section, based on the suggestions of the Reviewer #2.

Reviewer #2: Straight forward manuscript on a very interesting group of fungi. I have only minor revisions, mostly related to improvements for the discussion. Authors should focus on making the discussion more concise. Don't just repeat results, rather discuss how results improves understanding of Ampelomyces. Focus on the "including timing and quantifying mycoparasitism of Pseudoidium neolycopersici on tomato" portion of your title.

In the abstract, why not end with relevance in regards to developing a model system to further examine the use of Ampelomyces for biocontrol? Just emphasizing the strains origins from Japan or the use of digital microscopy is not quite novel, compared to the description of a model system to further study Ampelomyces mycoparasitism.

►Response to Reviewer #2

We thank Reviewer #2 for the review and for the helpful comments and corrections.

Minor comments were addressed in the RM (rephrasing, typo corrections) at all the positions marked by the Reviewer in the PDF file.

We added a sentence (P3 L49-50) to improve the ‘Abstract’ section in the RM, based on the suggestions of the Reviewer.

Based on the suggestion of the Reviewer, we shortened the discussion by removing sentences not directly related to the temporal characterization of the mycoparasitic process (P24 L470-P25 L495) in the RM.

We hope that the amendments on the text based on the helpful suggestions of the Reviewer will overcome the concerns described in the review.

Additional comments:

* This is more fitting of a general review; please be more concise and focus on how findings improves knowledge about Ampelomyces.

►Response

We revised the sentence (P24 L491-P25 L495) in the ‘Discussion’ section in the RM, based on the suggestions of the Reviewer.

Reviewer #3: This is a very interesting research. The phylogeny and mycoparasitic activity of Ampelomyces strains isolated from diverse powdery mildew hosts in Japan were explored. I recommend to accepting it for publishing. A few comments are shown in trace in the attached pdf file. 

►Response to Reviewer #3

We thank Reviewer #3 for the positive assessment of our manuscript, for the review and for the comments and corrections.

Minor comments were addressed in the revised manuscript (rephrasing, typo corrections) at all the positions marked by the Reviewer in the PDF file.

Additional comments:

* Please change the table format based on the author guide of PlosOne.

►Response

We changed the table format base on the author guide PLOS ONE.

* Please explain why you choose this strain.

►Response

This was one of the strains showing reliable and intensive in vitro sporulation. This information was added to the text (P14 L275-277 in the RM).

* Please check the symbol " after 1/2 correct or not.

►Response

We checked the symbol " after 1/2 and confirmed to be correct. 

* In Fig 2A, I think the main difference between mature and immature pycnidia is the size and not the color.

►Response

We amended the sentence (P19 L366-367 in the RM) to state that immature pycnidia are also smaller. The colour of the immature and the mature intracellular pycnidia is also very different as mentioned in the text.

---

## [Editor Report · Decision Letter 1]

27 Apr 2021

Ampelomyces strains isolated from diverse powdery mildew hosts in Japan: Their phylogeny and mycoparasitic activity, including timing and quantifying mycoparasitism of Pseudoidium neolycopersici on tomato

PONE-D-20-39222R1

Dear Dr. Teruo Nonomura,

We’re pleased to inform you that your manuscript has been judged scientifically suitable for publication and will be formally accepted for publication once it meets all outstanding technical requirements.

Kind regards,

Tofazzal Islam, Ph.D.

Academic Editor

PLOS ONE
---

## [Editor Report · Acceptance letter]

30 Apr 2021

PONE-D-20-39222R1 

*Ampelomyces* strains isolated from diverse powdery mildew hosts in Japan: Their phylogeny and mycoparasitic activity, including timing and quantifying mycoparasitism of *Pseudoidium neolycopersici* on tomato 

Dear Dr. Nonomura:

I'm pleased to inform you that your manuscript has been deemed suitable for publication in PLOS ONE. Congratulations! Your manuscript is now with our production department. 

Kind regards, 

on behalf of

Professor Dr. Tofazzal Islam 

Academic Editor

PLOS ONE